# Restoring Osteochondral Defects through the Differentiation Potential of Cartilage Stem/Progenitor Cells Cultivated on Porous Scaffolds

**DOI:** 10.3390/cells10123536

**Published:** 2021-12-14

**Authors:** Hsueh-Chun Wang, Tzu-Hsiang Lin, Che-Chia Hsu, Ming-Long Yeh

**Affiliations:** 1Department of Biomedical Engineering, National Cheng Kung University, Tainan City 70101, Taiwan; whc32002@hotmail.com (H.-C.W.); jeff.qbmc@yahoo.com.tw (T.-H.L.); 2Department of Orthopedic Surgery, National Cheng Kung University Hospital, College of Medicine, National Cheng Kung University, Tainan City 70403, Taiwan; persue630918@gmail.com; 3Medical Device Innovation Center, National Cheng Kung University, Tainan City 70101, Taiwan

**Keywords:** osteochondral tissue engineering, cartilage stem/progenitor cell, poly (lactic-co-glycolic acid) scaffold, migration, monophasic approach

## Abstract

Cartilage stem/progenitor cells (CSPCs) are cartilage-specific, multipotent progenitor cells residing in articular cartilage. In this study, we investigated the characteristics and potential of human CSPCs combined with poly(lactic-co-glycolic acid) (PLGA) scaffolds to induce osteochondral regeneration in rabbit knees. We isolated CSPCs from human adult articular cartilage undergoing total knee replacement (TKR) surgery. We characterized CSPCs and compared them with infrapatellar fat pad-derived stem cells (IFPs) in a colony formation assay and by multilineage differentiation analysis in vitro. We further evaluated the osteochondral regeneration of the CSPC-loaded PLGA scaffold during osteochondral defect repair in rabbits. The characteristics of CSPCs were similar to those of mesenchymal stem cells (MSCs) and exhibited chondrogenic and osteogenic phenotypes without chemical induction. For in vivo analysis, CSPC-loaded PLGA scaffolds produced a hyaline-like cartilaginous tissue, which showed good integration with the host tissue and subchondral bone. Furthermore, CSPCs migrated in response to injury to promote subchondral bone regeneration. Overall, we demonstrated that CSPCs can promote osteochondral regeneration. A monophasic approach of using diseased CSPCs combined with a PLGA scaffold may be beneficial for repairing complex tissues, such as osteochondral tissue.

## 1. Introduction

Articular cartilage has a limited capacity for self-repair and injury to the cartilage often progresses to osteoarthritis (OA) development [1]. Available medical interventions such as autologous chondrocyte implantation (ACI) [2], microfracture and mosaicplasty [3] can help to relieve symptoms but fail to produce functional cartilage. Recently, cell-based therapies for cartilage repair have mainly focused on chondrocytes [4], mesenchymal stem cells (MSCs) such as adipose derived stem cells [5] and bone marrow-derived stem cell [6], or tissue-specific progenitor cells [7]. Although chondrocytes exhibit excellent repair effects in cartilage tissue engineering, they are present in small amounts (less than 5%) in cartilage [8] and dedifferentiate in monolayer culture [9]. MSCs have been substituted for chondrocytes in osteochondral repair because of their rapid proliferation and multipotency characteristics [10]. However, the innate multilineage differentiation of MSCs [11] leads to the risk of hypertrophic growth [12] and endochondral ossification [13] in cartilage regeneration. Many strategies such as co-culture systems [14], oxygen pressure [15] and three-dimensional biomaterials [16] have been used for effective induction of chondrogenesis and stabilization on the differentiated chondrocyte phenotype from MSCs. In addition, new cell sources such as induced pluripotent stem cells (iPSC) or cartilage stem/progenitor cells (CSPC) have been exploited due to the limitless expansion in-vitro [17] or tissue-specific characteristic [18] in cartilage repair, respectively. However, tissue-specific progenitor cells possess both stem cell-like proliferative potential and tissue-specific phenotypes, thus the cells have gained attention and have been used to regenerate other tissues [19,20].

CSPCs were first identified on the surface of articular cartilage by Dowthwaite et al. [21]. These cells are located on one-third of the surface area of cartilage and also exist in the deep zone of cartilage [22], but only comprise 0.1–1% of the cartilage cell content. Much like MSCs, CSPCs have self-renewal and multilineage differentiation abilities [23]. Particularly, migration of CSPCs initiated by extracellular matrix (ECM) loss [24] and dead-cell debris [25] can prevent progressive cartilage loss [25].

Additionally, CSPCs have been proposed as a cell source for autologous transplantation in cartilage in equine models [26], and even in a pilot clinical trial in humans [18]. CSPCs also show better performance in neo-cartilage production in vitro than chondrocytes and MSCs in bioprinting [7]. Moreover, CSPCs from pathological joints exert immunomodulatory behavior in response to inflammatory stimulation [27]. Studies have demonstrated the therapeutic potential of CSPCs in cartilage repair and osteoarthritis [28,29]. However, the application of CSPCs combined with biomaterials and scaffolds have been explored in cartilage regeneration in vitro [7], but the effects and biological behavior in osteochondral repair in vivo have not been widely examined.

Multiphasic scaffolds are currently being developed to repair osteochondral tissue based on its heterogeneous, multilayered structure. Although multiphasic scaffolds resemble cartilage, calcified cartilage, and bone in osteochondral tissue, they may separate in vivo or even lead to poor osteochondral reconstruction [30]. However, monophasic scaffolds are fabricated from materials with a consistent porosity and overall stable architecture as well as one cell type. Monophasic approaches create a simple environment and are easy to manipulate for osteochondral tissue engineering and show potential for clinical use. In this study, we used a porous poly(lactic-co-glycolic acid) (PLGA) scaffold as a platform for cell encapsulation and three-dimensional (3D) culture which have been validated to promote the re-differentiation of chondrocytes and formation of the cartilage matrix [31].

Previous findings have demonstrated that the cells obtained from pathological joints, which contributed to tissue-specific therapeutic agents to improve cartilage repair [14,27]. In this study, we compared diseased CSPCs with IFPs to determine their self-renewal ability in culture, surface epitopes, and multi-differentiation potential. CSPC-laded on PLGA constructs were used as models of monolayered constructs to evaluate osteochondral regeneration in rabbit knees. We hypothesized that CSPCs combined with the monophasic, PLGA scaffold would strengthen the interface between cartilage and subchondral bone and enhance regeneration in osteochondral tissue in rabbit knees. Besides, the migration and path of CSPCs to the injury site for osteochondral regeneration were also evaluated.

## 2. Materials and Methods

### 2.1. Cell Isolation

Chondrocytes were isolated from human adult articular cartilage and analyzed. Adult articular cartilage samples (53–90-year-old subjects; mean, 70 years; *n* = 16) were dissected from non-lesion surface areas of the knee joints of patients without signs of rheumatoid involvement undergoing total knee replacement surgery. Patient consent was obtained, and the study protocol was approved on 6 March 2020 by the Institutional Review Board of National Cheng Kung University Hospital (No. A-ER-109-009). Primary chondrocytes were isolated from distal femoral condyles by enzymatic digestion. Briefly, articular cartilage tissue was cut into approximately 1 mm^3^ pieces and digested for 8 h at 37 °C in 0.2% (*w/v*) collagenase II (Sigma, St. Louis, MO, USA). Cells were transferred to a monolayer culture in Dulbecco’s modified Eagle’s medium supplemented with 10% fetal bovine serum (FBS; Thermo Fisher Scientific, Waltham, MA, USA) and penicillin/streptomycin (50,000 U/50 mg), and then cultured under standard conditions. CSPCs were isolated as previously described [23]. Briefly, 10-cm cell culture dishes were coated with fibronectin (Thermo Fisher Scientific, Waltham, MA, USA). Isolated full-depth chondrocytes were seeded onto the coated plates for 20 min at 37 °C in Keratinocyte-SFM (Gibco BRL, Grand Island, NY, USA) supplemented with the EGF-BPE (Gibco BRL, Grand Island, NY, USA), *N*-acetyl-L-cysteine, and L-ascorbic acid 2-phosphate sesquimagnesium salt (Sigma-Aldrich, Burlington, MA, USA) [32]. After 20 min, non-adherent cells, namely osteoarthritis chondrocytes (OACs), were removed. Adherent cells, namely CSPCs, were cultured until passage 3.

### 2.2. Colony Formation Analysis

One hundred cells were seeded into a 6 well plate and cultured with 10% FBS in low-glucose culture medium, and the medium was changed every 3 days. After 9 days, the cultures were fixed in 1% paraformaldehyde and stained with 1% crystal violet (Sigma-Aldrich, Burlington, MA, USA) in methanol for 10 min. All cell colonies with diameters of at least 2 mm were counted and their sizes were estimated.

### 2.3. Multilineage Differentiation

Osteogenesis, adipogenesis, and chondrogenesis were evaluated in CSPCs, IFPs.

For osteogenic and adipogenic differentiation, the cells were seeded at a density of 1 × 10^5^ cells per well in a 24-well culture plate until confluence and then the medium was changed to either MSC osteogenic differentiation medium (ScienCell Research Laboratories, Carlsbad, CA, USA) or MSC adipogenic differentiation medium (ScienCell Research Laboratories) with the addition of supplements accordingly at 37 °C in a 5% CO_2_ environment with regular medium changes for 21 days. Adipogenesis was observed by detecting lipid droplets via Oil Red staining and osteogenesis for mineralized bone matrix deposition by Alizarin Red S staining after 21 days.

For chondrogenic differentiation, 1 × 10^5^ cells were resuspended in 20 µL medium in individual wells of 24-well plates to perform high-density micromass cultures. The cultures were maintained for 2 h, and fresh medium was gently added for incubation for an additional 24 h. The medium was changed to MSC Chondrogenic differentiation medium (ScienCell Research Laboratories) with the addition of supplements accordingly at 37 °C in a 5% CO_2_ environment with regular medium changes for 21 days. Cells were then stained with Alcian blue to confirm the presence of glycosaminoglycans (GAGs) in chondrogenic differentiation. Stains were visualized with a light microscope (Olympus BX51, Tokyo, Japan).

### 2.4. Immunophenotype

Cultured cells in passage 3 were used flow cytometry analysis. The cells were suspended at 5 × 10^5^ cells/well and incubated with antibodies against surface markers; unstained cells were used as negative controls. *SOX9* was obtained from Spring Bioscience (Pleasanton, CA, USA), *DCX* was obtained from Invitrogen (Carlsbad, CA, USA), *CD44* was obtained from Novus Biologicals (Centennial, CO, USA), and others (*Type II, CD45, CD34, CD146, RUX2*) were obtained from Bioss (Woburn, MA, USA). Primary antibody-stained samples were incubated with a fluorescein isothiocyanate-conjugated rabbit anti-rabbit secondary antibody from Bioss (Woburn, MA, USA). A minimum of 50,000 events were evaluated with a FACS Calibur flow cytometer (BD Biosciences, San Jose, CA, USA). CellQuest Pro software (version 5.1) (BD Biosciences, San Jose, CA, USA) was used for further analysis.

### 2.5. Fabrication of Porous PLGA and CSPC/PLGA Scaffolds

The salt-leaching technique was used to fabricate the porous PLGA scaffold as previously described [33]. Briefly, a mixture of 20% PLGA chloroform solution with sodium chloride particles (300–500 μm in diameter) was poured into cylindrical molds and lyophilized for 1 day. The PLGA sponges were immersed into deionized water to dissolve the porogen. Finally, the cylindrical sponges (final dimensions were 3 mm in height and 3 mm in diameter) were formed by lyophilization. CSPCs were prepared at approximately 5.0 × 10^6^ cells/mL and seeded into PLGA scaffolds using a 0.43 mm syringe for 3D cultures. After 2 h, fresh medium was added before incubation for 1 day at 37 °C in a 5% CO_2_ environment.

### 2.6. Cell Tracking

To further track the bioactivity of the implanted CSPCs in vivo, CM-DiI (Molecular Probes, Eugene, OR, USA) was used as a non-targeted probe for CSPCs before transplantation. All procedures were performed as previously described [34]. The applied label, CM-Dil (excitation: 553 nm; emission: 570 nm), for CSPCs was monitored by red fluorescence in the defect zones at 4 and 12 weeks after implantation. The Xenogen IVIS^®^ Spectrum Noninvasive Quantitative Molecular Imaging System (PerkinElmer, Waltham, MA, USA) was used for optical imaging for cell tracking.

### 2.7. Animal Procedures

This study was performed in accordance with protocols approved by the Institutional Animal Care and Use Committee of National Cheng Kung University (No. 106163).

All surgical procedures were similar to those described previously [34]. New Zealand White male rabbits (4–5 months old; Livestock Research Institute, Taiwan) weighing 2–3 kg were used in this study. A full-thickness osteochondral defect (3 mm in diameter and 3 mm in depth) was made in the center of the medial femoral condyle [35] by using an electric drill. The rabbits were allocated randomly into four groups: empty defect (ED) (*n* = 12), sham (*n* = 4), PLGA scaffold (*n* = 12), and CSPC/PLGA (*n* = 12) for the osteochondral defect. The schematic diagram of the study design is shown in (Appendix A). The PLGA scaffold was inserted into the defect hole by press-fitting. In the CSPC/PLGA groups, the cells were seeded into PLGA on the day before surgery. The remaining surgical procedures were the same as those used in the PLGA group. Postoperatively, the animals were returned to their cages and allowed free cage activity without immobilization. The rabbits were euthanized after 4 or 12 weeks via intravenous injection of 2meq/kg KCL (Taiwan Biotech, Taoyuan, Taiwan) and the repaired osteochondral tissues were harvested for further examination.

### 2.8. Macroscopic Evaluation

The rabbits were euthanized postoperatively 4 and 12 weeks, and their knees were harvested. There were no redness, swelling around the knee joints in all rabbits. The regenerated tissue was scored for their gross morphology according to a modified Wayne’s grading scale [36] (Appendix A). Macroscopic scores were assessed blindly by two investigators.

### 2.9. Micro-CT Evaluation

All procedures used for micro-computed tomography (CT) analysis were described previously [34]. Briefly, harvested femoral condyles were analyzed by a high-resolution micro-CT 1076 scanner (Skyscan, Kontich, Belgium). The defect region was determined and a cylindrical region of interest 3 mm in diameter × 3 mm deep was restricted to assess subchondral bone healing qualitatively and quantitatively in different groups. Repair was determined as the percentage bone volume over total volume (% BV/TV) and the width of the bone growth as trabecular thickness (Tb.Th).

### 2.10. Histological and Immunohistochemical Processing

Histological sections were prepared by the Department of Pathology at National Cheng Kung University Hospital, Tainan, Taiwan. The resected femurs underwent standard processing, including 10% neutral-buffered formalin fixation, gradient dehydration, decalcification, sectioning perpendicular to the longitudinal axis, infiltration, and paraffin embedding. Sections (4 μm thick) were stained with hematoxylin and eosin for general observation, Masson’s trichrome staining for total collagen content and alignment, and Safranin-O staining for GAG synthesis; the sections were examined by light microscopy (Olympus BX51, Tokyo, Japan) and recorded using a digital CCD camera (Olympus DP70, Tokyo, Japan). Immunohistochemistry was performed to detect the expression of type I and II collagen and observe regeneration in the osteochondral defect. Endogenous peroxidase was treated with peroxidase-blocking reagent included in the Rabbit/Mouse HRP-DAB detection system (BioSB, Santa Barbara, CA, USA) for 10 min, and then the samples were boiled in citrate buffer for 15 min for epitope retrieval. The Rabbit/Mouse HRP-DAB Polymer detection system was used as the secondary antibody at room temperature for 20 min. Finally, the signal was identified as a brown precipitate using 3,3′diaminobenzidine (DAB) substrate (BioSB Santa Barbara, CA, USA). The samples were counterstained with hematoxylin (BioSB Santa Barbara, CA, USA), and the slides were dehydrated and cover-slipped.

### 2.11. Statistical Analysis

Data are expressed as the mean ± standard error of the mean. Because the data were not normally distributed, nonparametric statistics were used for analysis. Statistical analyses were performed using SPSS v. 17.0 software (SPSS, Inc., Chicago, IL, USA). Kruskal-Wallis one-way analysis was used for comparisons between groups. The generalized estimating equations was performed to evaluate data from different time-points [37]. A *p* value of <0.05 was considered to indicate statistical significance.

## 3. Results

### 3.1. Characterization of CSPCs after Isolation and Cultivation In Vitro

#### 3.1.1. Assessment of CSPC Attachment and Spreading

One day after seeding, CSPCs (P0) (Figure 1A) were observed under a light microscope. CSPCs emerged as colonies and gradually extended from the center to the edge of the plate. The shape of CSPCs (P1) (Figure 1A) became spindly-like stem cells and grew rapidly in the culture dish (Figure 1A).

#### 3.1.2. Colony Formation Analysis

CSPCs derived from patients with OA formed colonies (Figure 1B,C) and possessed a similar colony-forming ability as IFPs, but with a different conformation. CSPCs were prone to concentrate to form a 3D structure, and thus the diameter of the colonies was larger than 2 mm. In contrast, IFPs were accustomed to 2D culture conditions, and thus some colonies were less than 2 mm. The colony diameter was larger in CSPCs than in IFPs, but more colonies were formed from IFPs than from CSPCs. Colonies in each dish were counted after staining with crystal violet (Figure 1B,C), revealing no significant difference between CSPCs and IFPs in colony-forming efficiency (92.25 ± 5.64%, 90.25± 1.65%, respectively, *p* = 0.21).

#### 3.1.3. Multilineage Differentiation

To further assess the stem cell characteristics of CSPCs, the multilineage differentiation potential was evaluated. CSPCs and IFPs underwent induced differentiation into osteogenic, adipogenic, and chondrogenic lineages (Figure 1D) after 21 days, showing positive responses.

#### 3.1.4. Immunophenotype Assay of CSPCs

Low cytometry (Figure 2A) showed that CSPCs were positive for well-recognized MSC-associated surface markers (*CD90* and *CD44*), whereas hematopoietic stem cell-associated markers (*CD34*, and *CD45*) exhibited low expression. CSPCs showed moderate expression of intracellular protein (*collagen type II*) and chondrogenesis and osteogenesis transcription factors (*SOX9, RUNX2*) and high expression of a CSPC-associated surface marker (*CD146*). However, CSPCs also exhibited intermediate expression of doublecortin (*DCX*). In contrast, OACs displayed negative expression of *RUNX2* (Figure 2B). These results indicated that CSPCs possess similar epitope profiles as MSCs, chondrocytes, and osteoblasts.

### 3.2. Morphology of CSPCs on PLGA Scaffolds

We used the salt-leaching technique to fabricate PLGA scaffolds 3 mm in diameter and 3 mm in height, as shown in Figure 3A. The interior pore structure and morphology of the PLGA scaffolds were clearly observed by scanning electron microscopy. The average pore sizes of the PLGA scaffolds were 300–500 μm and were controlled by the size of sodium chloride porogen (Figure 3B,C). The porosity was over 90%, as demonstrated previously [34].

To evaluate the biocompatibility of PLGA scaffolds to support chondrogenic differentiation of CSPCs, CSPCs were seeded into PLGA scaffolds and cultured for 7 days. After 7 days, CSPCs adhered to the surface of PLGA, further demonstrating the growth and proliferation of CSPCs on the interior surface of PLGA scaffolds (Figure 3D,E).

### 3.3. Location and Biological Activity of CSPCs Evaluated by In Vivo Imaging System (IVIS) and Spectrum CT Analyses In Vivo

The bioluminescence value of the CSPC/PLGA group was approximately 3.8 × 10^8^ photons/s/cm^2^/steradian (p/s/cm^2^/sr) after 4 weeks. Bioluminescence in the defect site was clear at 4 weeks after implantation in the CSPC/PLGA group (Figure 4A).

However, bioluminescence was still detectable on the cartilage surface and appeared to diffuse out of the defect sites in the CSPC/PLGA group after 12 weeks (Figure 4A). The bioluminescence was remained bright, with a value of approximately 3.6 × 10^8^ p/s/cm^2^/sr. Bioluminescence was not detected at 4 or 12 weeks in the PLGA group.

We also examined the depth of CSPC migration in vivo. After 4 weeks, CSPCs had spread evenly into the PLGA scaffold. Interestingly, after 12 weeks, CSPCs centralized to the subchondral bone and close to unrepaired sites (Figure 4B). Fluorescence was not detected in the control group (unstained).

### 3.4. Macroscopic Observations and Quantitative Scores

#### 3.4.1. Gross Appearance

No inflammatory reactions or joint contractures were found throughout the postoperative period in any group. After 4 weeks, the PLGA scaffold did not appear to be fully degraded in both the PLGA and CSPC/PLGA groups. The color of the repaired tissue in the CSPC/PLGA group was more similar to that of the host tissue than that of the PLGA group after 12 weeks. Concave areas in the injured regions were found in the empty defect (ED) group (Figure 5A).

#### 3.4.2. Quantitative Scores

At 4 weeks, the total scores in the CSPC/PLGA (9.5 ± 0.65) and PLGA (5.25 ± 0.48) groups were significantly different (*p* < 0.01) and both significantly higher than those of the ED group (1.0 ± 0.7) (*p* < 0.01, for both) (Figure 5B).

At 12 weeks, the score of the CSPC/PLGA group (11.5 ± 0.5) was significantly higher than those of the ED was (4.25 ±1.44) and PLGA (7.75 ± 0.48) groups (*p* < 0.01, *p* < 0.01, respectively) (Figure 5B). The score of the PLGA group was also significantly higher than that of the ED group (*p* = 0.03) (Figure 5B).

### 3.5. Micro-CT Analysis

#### 3.5.1. Findings after 4 Weeks

A newly formed osseous matrix emerged and grew from the edge to central area of the defect in the PLGA and CSPC/PLGA groups at 4 weeks after implantation (Figure 6A). The BV/TV and Tb.Th values of the CSPC/PLGA (15 ± 0.58, *p* < 0.01; 0.14 ± 0.005, *p* < 0.01, respectively) and PLGA (13 ± 0.58, *p* = 0.035; 0.12 ± 0.005, *p* = 0.008, respectively) groups were significantly different from those of the ED group (11 ± 0.58; 0.093 ± 0.004, respectively) (Figure 6B,C). There were significant differences between the CSPC/PLGA (15.0 ± 0.58, *p* = 0.035; 0.14 ± 0.005, *p* = 0.027) and PLGA groups (13.0 ± 0.58, 0.12 ± 0.005, respectively) in BV/TV and Tb.Th values (Figure 6B,C). Significant differences in BV/TV and Tb.Th values were observed in every group except for the sham group (Figure 6B,C).

#### 3.5.2. Findings at 12 Weeks

At 12 weeks after implantation, a newly synthesized mineral matrix filled up the defect site in the CSPC/PLGA group and exhibited the highest BV/TV and Tb.Th values (24.52 ± 1.26, 0.20 ± 0.014, respectively); both values were significantly higher than those of the PLGA (21 ± 0.57, *p* = 0.03; 0.175 ± 0.003, *p* = 0.048, respectively) and ED (16.33 ± 0.88, *p* = 0.003; 0.14 ± 0.005, *p* = 0.007, respectively) groups. However, the BV/TV values of the CSPC/PLGA group were significantly lower than those of the sham group were (24.52 ± 1.26; 35.33 ± 0.33, *p* < 0.01), whereas the Tb.Th values (0.20 ± 0.014, 0.203 ± 0.003) were not significantly different from those in the sham group (Figure 6B,C).

#### 3.5.3. Comparison by Micro-CT Analysis at 4 and 12 Weeks

We compared the results obtained after 4 and 12 weeks. The CSPC/PLGA group showed significant differences in the BV/TV (15 ± 0.58, 24.52 ± 1.26, respectively, *p* = 0.001) and Tb.Th (0.14 ± 0.005, 0.20 ± 0.014, respectively, *p* = 0.007) values over time. There were also significant differences in the PLGA group (BV/TV: 13 ± 0.58, 21± 0.57, respectively, *p* < 0.01; Tb.Th: 0.12 ± 0.005, 0.175 ± 0.003, respectively, *p* < 0.01) (Figure 6B,C).

### 3.6. Histology

No inflammatory responses were found at the transplantation sites in our in vivo experiments. At week 4, smooth cartilage surfaces were only observed in the CSPC/PLGA group. Although undegraded PLGA remained in the subchondral bone, regenerated tissue was continuous and well integrated with the host tissue in the cartilage. The cartilage and subchondral bone were easily distinguished by Masson’s trichrome staining in the CSPC/PLGA group, whereas there was a discontinuous cartilage surface and disorganized regenerated tissue in subchondral bone in the PLGA group. Furthermore, the CSPC/PLGA group exhibited abundant GAG deposition as shown by Safranin O staining. Large amounts of fibrocartilage and inadequate reparative tissue filled the defect site in the ED group (Figure 7).

After 12 weeks, the PLGA scaffolds were almost degraded and new tissue had grown in to substitute the PLGA scaffolds in the PLGA and CSPC/PLGA groups. Smooth surfaces, adequate cartilage thickness, rich GAG formation, and well-aligned cells were observed in the CSPC/PLGA groups. However, there were still some empty spaces in the subchondral bone in the CSPC/PLGA groups. In the PLGA group, concave surfaces were still present in the middle of the defect site. However, bone formation in the defect site at 12 weeks was more mature than that at 4 weeks. The ED group samples showed high levels of fibrous tissue formation, with scarce hyaline cartilage and no tissue in the subchondral bone (Figure 7).

We analyzed the regenerated tissue by immunohistochemistry to assess the levels of type II collagen (*COLII*) and type I collagen (*COLI*). At 4 weeks, *COLI* and *COLII* were both present in the entire defect site in both groups, particularly in the CSPC/PLGA group. The surface of cartilage was regenerated first and exhibited *COLII* expression, which extended down to subchondral bone and the shape of PLGA was apparent in the CSPC/PLGA group. In contrast, cartilage and subchondral bone were slightly regenerated and expressed both *COLII* and *COLI* in the PLGA group. At 12 weeks, the cartilage and subchondral bone were regenerated progressively but expressed *COLII* and *COLI* in the PLGA group. In contrast, *COLII* expression was visible in the smooth cartilage in the CSPC/PLGA group (Figure 8).

## 4. Discussion

In this study, we demonstrated the potential of human cartilage stem/progenitor cells from patients with OA as a novel cell source for osteochondral regeneration. The CSPCs harvested and sorted from patients with late-stage OA did not dedifferentiate like chondrocytes but expressed colony-forming ability and multilineage differentiation. To identify the surface antigens of CSPCs, high levels of the markers of CSPCs (*CD146*), MSC-associated surface markers (*CD90*) and marker of joint-resident MSCs (*CD44)* [38] were expressed, and the level of hematopoietic stem cell-associated markers (*CD34*, and *CD45*) were decreased. CSPCs not only expressed intracellular protein (*type II*), but also chondrogenesis and osteogenesis transcription factors (*SOX9, RUNX2*). However, *DCX* was also observed by flow cytometry analysis. *DCX* is only expressed in human and mouse articular chondrocytes and not in endochondral chondrocytes [39]. Once MSCs differentiate into endochondral chondrocytes, the expression of *DCX* is lost [40]. Emerging expression of *DCX* indicated that CSPCs could regenerate functional hyaline cartilage. This type of cartilage differs from endochondral cartilage which undergoes terminal differentiation [41]. Moreover, these results suggest that CSPCs can differentiate into chondrocytes or osteoblast lineage cells. However, CSPCs remain distinct from these cells.

In this study, we used the salt-leaching method to fabricate a 3D PLGA scaffold. This procedure has advantages such as the ability to control pore size, mechanical rigidity, and degradation rate according to the PLA-to-PGA ratio. Using this procedure, CSPCs can lie firmly in the pores of the PLGA structure during daily activity. Previous studies demonstrated that the degradation half-time of PLGA scaffolds in vitro is 3–4 weeks [42] with complete degradation occurring by 12 weeks. The architecture of the PLGA scaffold was fixed to the defect without using periosteum as a cover, and the cells were easily inoculated without leakage. The structure also provided a topographical cue to promote cell migration, attachment, and proliferation, as well as tissue regeneration.

Retaining cell viability after implantation is key for the consequent repair in cellular repair approaches. To confirm that the CSPCs were alive and localized within the defects after 12 weeks, CM-DiI was used to trace the transplanted CSPCs. CSPCs remained firmly and incorporated into the PLGA scaffold after 12 weeks, whereas CSPCs were absent from the PLGA group. Thus, CSPCs participated in the entire regenerative process. To further investigate the possible behavior of CSPCs in vivo, we traced the migration potential of CSPCs. CSPCs migrated down to the subchondral bone and concentrated at the reparative sites after 12 weeks. Gerter et al. also showed that CSPCs can penetrate 1000–1400 μm deep into cartilage or even 1700 μm in vitro [28]. CSPCs respond to injury, express *Lubricin* (*proteoglycan 4*), and resurface the articular cartilage in early OA [25]. However, CSPCs appear to migrate throughout the articular cartilage in response to injury during late OA [43]. Our study confirms the role of CSPCs in OA progression. CSPCs manipulated in vitro were still attracted by chemokine factors and were viable for up to 12 weeks; CM-Dil staining of CSPCs deep in the osteochondral tissue indicated their ability to proliferate. Moreover, we further preliminary showed that extracellular vesicle-derived medium from CSPCs significantly enhanced the proliferation of both chondrocytes and osteoblasts compared to IFPs (Appendix A). This indicates that CSPCs possessed bioactivity and the potential to regenerate osteochondral tissue even after manipulated in vitro.

According to the histology results, at 4 weeks after transplantation, the CSPC/PLGA group exhibited smooth cartilage surface and the effect extended from cartilage to bone after 12 weeks. Upon involvement of CSPCs, higher levels of GAG and collagen synthesis, good cell alignment, and good integration with host tissue were observed. This indicates that the migration of implanted CSPCs enhanced tissue integration, and CSPCs in the host tissue responded vigorously to *SDF-1**α* [25], a key chemokine that regulates stem cell migration and homing to sites of tissue damage and contributes to regeneration overall. This is consistent with a study by Lu et al., who demonstrated that cell migration at the interface of engineered cartilage and surrounding cartilage results in stronger host-graft tissue integration [44].

The CSPC/PLGA group showed a smooth cartilage surface and abundant GAG formation, which was well integrated with the host tissue after 12 weeks. However, the repair of subchondral bone was gradual and remained incomplete after 12 weeks in the CSPC/PLGA group. The CSPC/PLGA group showed a similar Tb.Th value to that of the sham group, but not the BV/TV ratio in micro-CT data after 12 weeks. These results are unsatisfactory, and we predicted that CSPCs may undergo intramembranous ossification rather than endochondral ossification during bone formation. The expression of *DCX* and *RUNX2* of CSPCs confirmed this hypothesis. *DCX* is only expressed in articular chondrocytes and disappeared in endochondral ossification. Otherwise, *RUNX2* was expressed to trigger osteogenesis. Second, we traced the behavior of CSPCs in vivo. We found CSPCs responded to injury and first resurfaced the cartilage, after which CSPCs migrated and participated in communication between the articular cartilage and subchondral bone [45]. Uncontrolled cell migration may accelerate tissue disruption and slow ECM production. The whole process is similar to the CSPC distribution during OA pathogenesis [29]. However, adding bone morphogenetic protein 6 [43] to block cell migration and enhance ECM production may accelerate bone regeneration.

In an investigation done previously [34], we used continuous passive motion (CPM) to promote and maintain the chondrogenesis in endothelial progenitor cells (EPCs) loaded PLGA scaffolds during osteochondral regeneration [34]. In the current study, CSPCs migrated and attracted additional cells to repair osteochondral tissue without other intervention in PLGA Scaffolds. However, a combination of CSPCs and EPCs with biphasic scaffold might benefit heterogeneous osteochondral tissue at early disease stage and encourage sufficient endogenous cell-based repair attempts.

The whole mechanism of repair in the osteochondral in CSPC/PLGA group was predicted as follows: cultured CSPCs seeded in PLGA implanted at the osteochondral defect and spread evenly in the PLGA scaffold, with some resident CSPCs in the host cartilage. After the construct had been implanted for some time, resident CSPCs were attracted into the PLGA scaffold by injury, and most cultured CSPCs gathered on the upper layer of the PLGA scaffold, with few cultured CSPCs spread in the rest of the PLGA scaffold. When new cartilage was formed, cultured CSPCs were attracted by injury and migrated to the unrepaired site in the subchondral bone (Figure 9).

## 5. Conclusions

Cartilage samples were collected from patients with OA after TKR, which would normally be disposed of as waste in the clinic. We demonstrated that human diseased CSPCs possess migration ability and promote adequate osteochondral regeneration in rabbits. CSPCs from xenogeneic, allogeneic, or even autologous species can be utilized in this attractive approach for cartilage defect repair. Furthermore, the migratory potential of CSPCs can improve cell recruitment into cartilage defects without perforating the subchondral bone plate. This strategy can also be used to treat partial-thickness cartilage defects.

Some CSPCs were originally retained in the cartilage but were isolated and enriched during expansion. These cells exhibited therapeutic effects. Thus, CSPCs from diseased joint show potential for manufacturing cell-based products for clinical utilization. CSPCs combined with a PLGA scaffold, a monophasic approach, may be useful for regenerating complex tissues such as osteochondral tissue at an early disease stage and be useful for cell-based repair.

## Figures and Tables

**Figure 1 cells-10-03536-f001:**
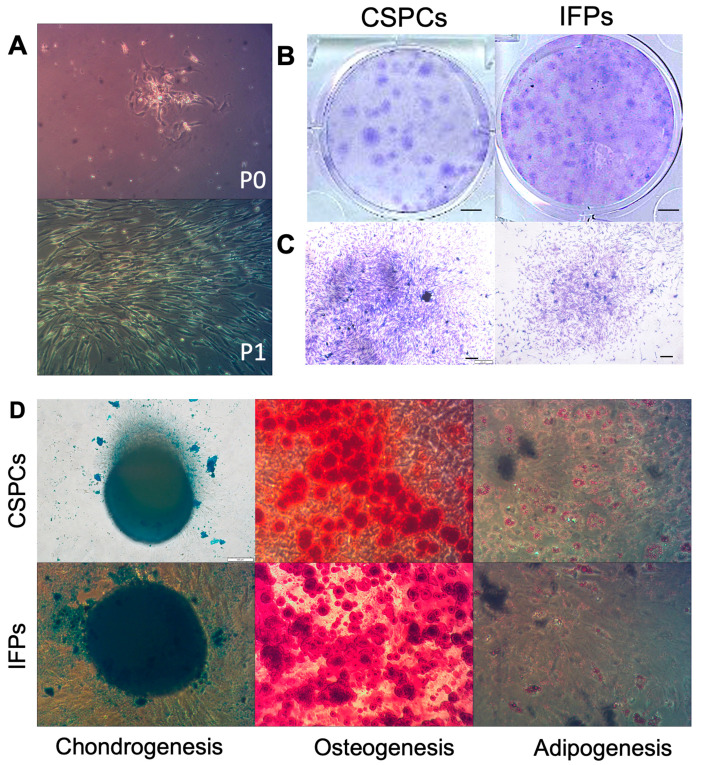
(**A**) Cell morphology of two-dimensional cultures. (**B**) Colony-forming ability of CSPCs of IFPs after 9 days of culture, scale bar = 5mm. (**C**) Magnification of (**B**), scale bar = 200 μm. (**D**) Multilineage differentiation potential of CSPCs and IFPs after 21 days, scale bar = 100 μm.

**Figure 2 cells-10-03536-f002:**
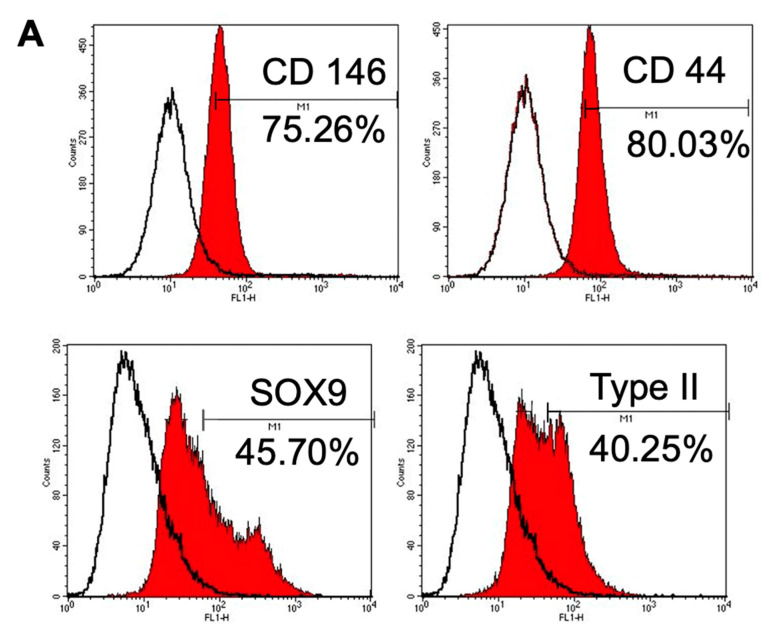
Flow cytometric analysis of CSPCs and OACs. (**A**) Representative flow cytometry analysis of CSPCs and (**B**) OACs.

**Figure 3 cells-10-03536-f003:**
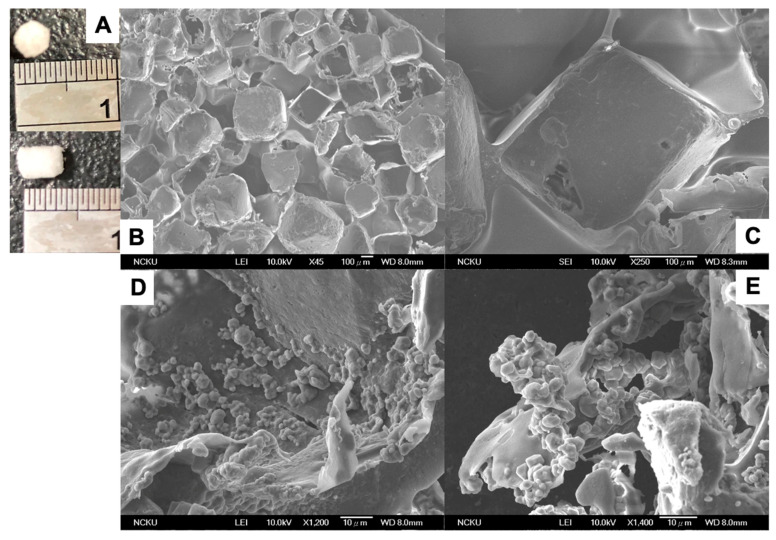
Characteristics of PLGA scaffolds and morphology of CSPCs seeded on PLGA scaffolds and cultured for 7 days. (**A**) PLGA sponge scaffold; various scanning electron microscopy images of PLGA structure at magnifications of (**B**) ×45 and (**C**) ×250, and morphology of CSPC-seeded PLGA scaffolds at magnifications of (**D**) ×1200 and (**E**) ×1400.

**Figure 4 cells-10-03536-f004:**
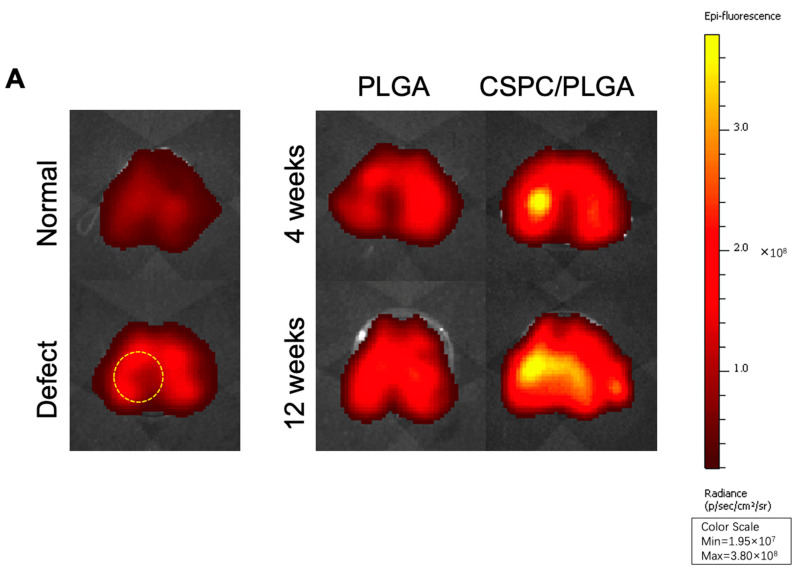
(**A**) Location of CSPCs after transplantation in the normal and defect-only groups. The yellow dotted circle indicates the defect site. Localization of CSPCs by CM-Dil at 4 and 12 weeks after PLGA scaffold transplantation. Bioluminescence was determined with an in vivo imaging system in the dissected tissue of the femurs of the CSPC/PLGA groups. (**B**) Migration potential of CSPCs in CSPC/PLGA at 4 and 12 weeks after implantation, scale bar: 200 μm.

**Figure 5 cells-10-03536-f005:**
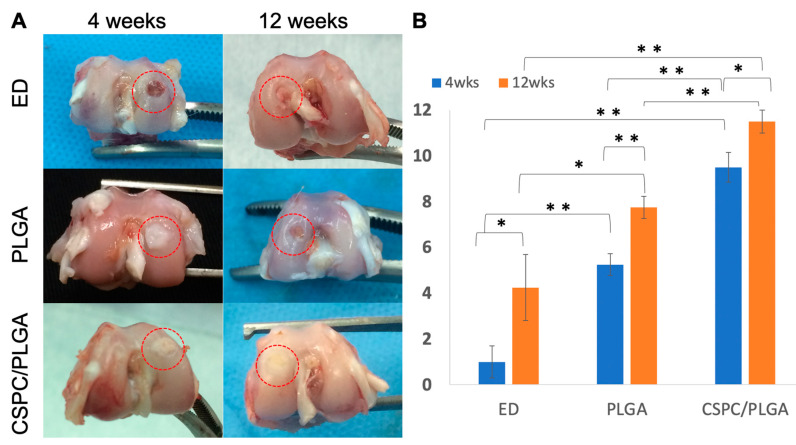
Gross appearance of articular cartilage defects at 4 and 12 weeks post-operation. Red dotted circles indicate defect sites (**A**). Qualitative scores of the gross appearances of the empty defect (ED), PLGA, and CSPC/PLGA groups at 4 and 12 weeks post-operation (**B**). * significant within-group difference (*p* < 0.05). ** significant within-group difference (*p* < 0.01).

**Figure 6 cells-10-03536-f006:**
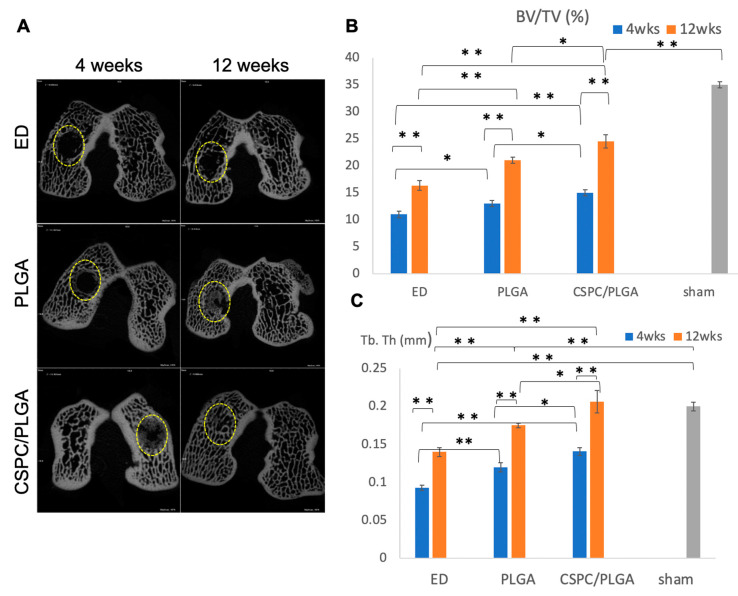
Bone regeneration over time. (**A**) Bone assessment of 2D micro-CT images. The yellow dotted circles indicate the defect sites. (**B**) Ratio of bone volume to tissue volume (BV/TV). (**C**) Thickness of trabecular bone (Tb.Th). * significant within-group difference (*p* < 0.05). ** significant within-group difference (*p* < 0.01).

**Figure 7 cells-10-03536-f007:**
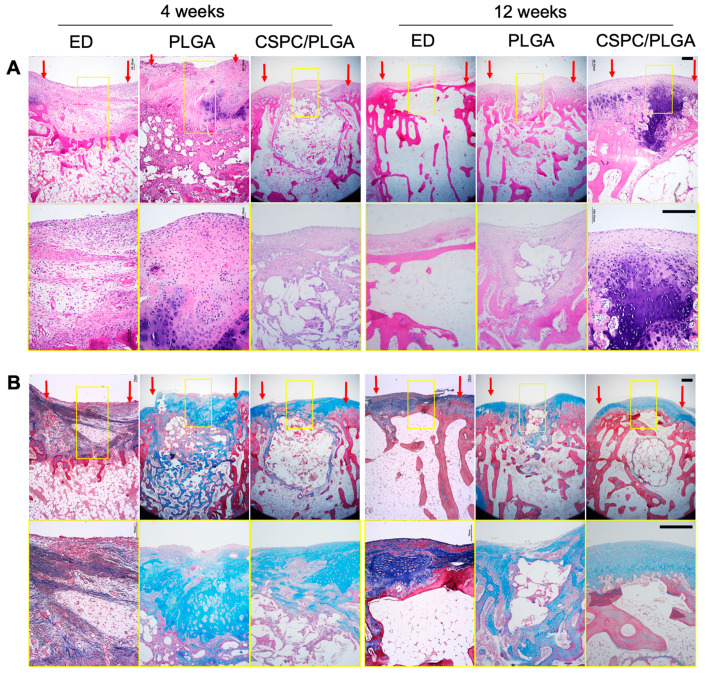
Histology of the different groups. Representative images of histological examinations by (**A**) Hematoxylin and eosin staining, (**B**) Masson’s trichrome, and (**C**) Safranin O staining, magnification: 4×. Yellow squares denote the magnified area, magnification: 10×. Red arrows show the border of the repaired tissue, scale bars: 500 μm.

**Figure 8 cells-10-03536-f008:**
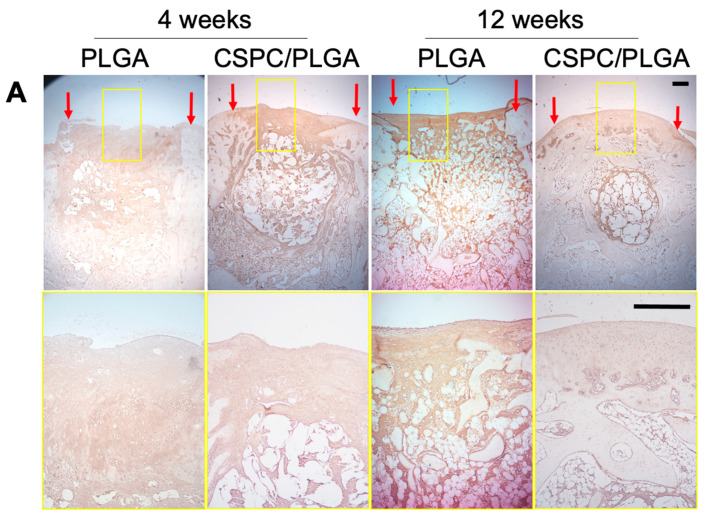
Representative images of specific proteins within the defect sites detected by immunohistochemistry (IHC) staining. (**A**) *COLI* and (**B**) *COLII*, magnification: 4×. Red arrows show the border of the repaired tissue. Yellow squares denote the magnified area, magnification: 10×, scale bars: 500 μm.

**Figure 9 cells-10-03536-f009:**
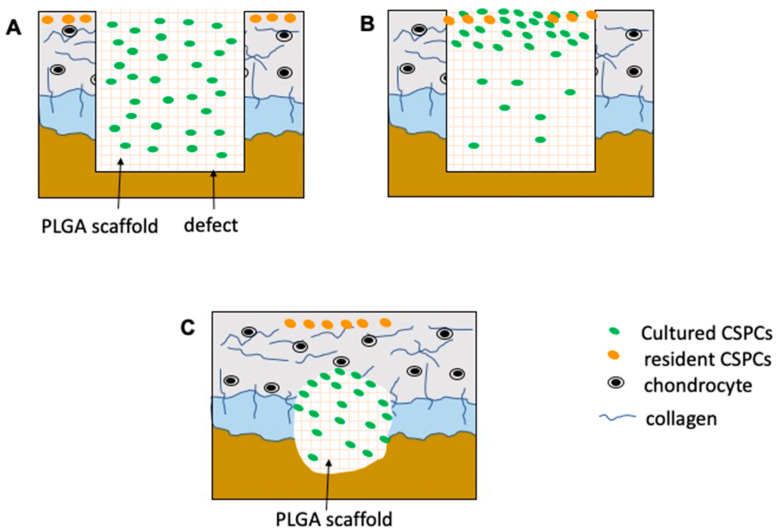
Predicted pathway of CSPCs combined PLGA scaffold in osteochondral regeneration. (**A**) CSPC/PLGA construct was implanted into osteochondral defect. (**B**) Resident CSPCs migrated into the edge of CSPC/PLGA construct. (**C**) Cultured CSPC migrated to the unrepaired site, and resident CSPCs resurfaced the cartilage.

## Data Availability

Data can be requested from the corresponding author.

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
