# Peer review of "Restoring Osteochondral Defects through the Differentiation Potential of Cartilage Stem/Progenitor Cells Cultivated on Porous Scaffolds"

_cells, 2021, doi:10.3390/cells10123536_

Round 1
Reviewer 1 Report
Wang et al. evaluated the characteristics and potential of human CSPCs combined with PLGA scaffolds to induce osteochondral regeneration in preclinical in vitro and in vivo models. The study could be interesting but numerous points and details need to be provided. Furthermore, the discussion should be considerably improved and should not speculate on the functions that the cells could have and, on the superiority, (or similarity) over MSCs.
INTRODUCTION
Lines 33-37: Please insert appropriate references for each sentence.
Line 34: “Available medical interventions…” Please briefly mention at least the most important ones.
Lines 35-37: “Recently, cell-based therapies for cartilage repair have mainly focused on chondrocytes, mesenchymal stem cells (MSCs), or tissue-specific progenitor cells”. Can you specify which mesenchymal cells you are referring to? Bone marrow MSCs, ADSC or others?
Line 41: please substitute “inherent” with physiological.
Line 41-43: “The inherent differentiation ability of MSCs leads to the risk of hypertrophic growth, terminal differentiation, and subsequent tissue calcification.” Please clarify this sentence.
Line 60: “…combined with biomaterial constructs” Please replace with “biomaterials and scaffolds”.
Line 61: “…have not been widely examined.” What has been examined to date?
Line 74: “…CSPC-laden PLGA” Did you mean CSPC-loaded on PLGA?
Lines 80-81: Please eliminate this sentence from the introduction.
MATERIALS AND METHODS
Lines 100-101: “After 20 min, non-adherent cells, namely osteoarthritis chondrocytes (OACs)…” Why OA chondrocytes? Previously it is written that articular cartilage samples were dissected from non-lesion surface areas of the knee joints. I can't understand the name of osteoarthritis chondrocytes.
Line 106: Why did you first use FBS 20% and then 10%?
Lines 119-120. “Adipogenesis was observed by detecting lipid droplets via Alizarin Red S staining and osteogenesis for mineralized bone matrix deposition by Oil Red O staining after 21 days.” It is the reverse, i.e. adipogenesis was observed by detecting lipid droplets by Oil Red O staining while osteogenesis for mineralized bone matrix was detected by Alizarin Red S.
Lines 122-129: Did you make micromasses in wells? Only by resuspending? No centrifuge?
Lines 132-141: The only way to reliably characterize MSCs through surface markers is for the cells to be cells negative for surface markers such as CD45, CD34, CD14 or CD11b, CD79 or CD19 and HLA-DR, but positive for CD105, CD73 and CD90.
Lines 163-177:
-Did you do a power analysis to establish the number of animals per group?
- Method of animal euthanasia?
-Please replace sacrificed to euthanized.
- Have you evaluated and removed the popliteal lymph nodes?
Lines 186-192: why didn't you do also the trabecular thickness and separation.
RESULTS
Line 221: Figures 1A and B are of low resolution, and it is not possible to see what is described in text.
Line 228: As for Figures 1A and B, also Figures B and C are of low resolution, and it is not possible to understand what is described in text.
Line 240: Figure 1D are of low resolution and described in the text in a superficial way.
Line 248. The immunophenotypes of CSPCs were poorly analyzed. What about ACAN, ADIPOQ, ALP, BGLAP, PPARG?
Line 349: The histological images are too many, too small and with too low a magnification to evaluate what is reported in the text.
Line 374: The immunohistochemical images all look the same and the staining appears nonspecific. To evaluate what is reported in the text or in any case to detect differences, larger magnifications are required, i.e. 20x, 40x, 80x.
Figure 7 and 8: please report the magnifications for each image.
DISCUSSION
Please do not mention the figures in the discussion section.
‘The CSPCs harvested and sorted from patients with late-stage OA did not dedifferentiate like chondrocytes and behaved similarly to MSCs in terms of colony-forming ability and multilineage differentiation’. The data obtained are too premature to state that CSPCs behaved similarly to MSCs. I think that CSPCs could represent a novel cell source for osteochondral regeneration, but supplementary studies and analyses are mandatory. Few differentiating factors and few surface markers were analyzed as well as too few immunohistochemical evaluations.
Unfortunately considering that many images are low resolution, the comments present in the discussion are difficult for me to understand and comment on.
I think that the discussion should be less peremptory, perhaps assuming specific CSPCs mechanisms rather than giving them as certain.
Author Response
Dear Editor:
Thank you for providing us the opportunity to submit a revised version of our manuscript and to reviewers for their invaluable comments. We have gone through the reviewer’s feedback and modified the original manuscript as recommended. Our point-by-point responses are shown below.
Reviewer 1:
Comments and Suggestions for Authors
Wang et al. evaluated the characteristics and potential of human CSPCs combined with PLGA scaffolds to induce osteochondral regeneration in preclinical in vitro and in vivo models. The study could be interesting but numerous points and details need to be provided. Furthermore, the discussion should be considerably improved and should not speculate on the functions that the cells could have and, on the superiority, (or similarity) over MSCs.
Author response: Thank you for the suggestion. We have corrected the sentence to “ The CSPCs harvested and sorted from patients with late-stage OA did not dedifferentiate like chondrocytes but expressed colony-forming ability and multilineage differentiation” in the revised manuscript.
INTRODUCTION
Lines 33-37: Please insert appropriate references for each sentence.
Author response: Thank you for the suggestion. We have inserted appropriate references for each sentence in the revised manuscript.
Line 34: “Available medical interventions…” Please briefly mention at least the most important ones.
Author response: Thank you for the suggestion. We have mention “autologous chondrocyte implantation (ACI), microfracture and mosaicplasty” in the revised manuscript.
Lines 35-37: “Recently, cell-based therapies for cartilage repair have mainly focused on chondrocytes, mesenchymal stem cells (MSCs), or tissue-specific progenitor cells”. Can you specify which mesenchymal cells you are referring to? Bone marrow MSCs, ADSC or others?
Author response: Thank you for the suggestion. We have specified adipose derived stem cells and bone marrow-derived stem cells in the revised manuscript.
Line 41: please substitute “inherent” with physiological.
Author response: Thank you for the suggestion. We have substituted “inherent” to “innate” in the revised manuscript.
Line 41-43: “The inherent differentiation ability of MSCs leads to the risk of hypertrophic growth, terminal differentiation, and subsequent tissue calcification.” Please clarify this sentence.
Author response: Thank you for your question. We have clarified this sentence as below “The innate multilineage differentiation of MSCs [11] leads to the risk of hypertrophic growth [12] and endochondral ossification [13] in cartilage regeneration.” And we also added the appropriate references in sentence.
Line 60: “…combined with biomaterial constructs” Please replace with “biomaterials and scaffolds”.
Author response: Thank you for the suggestion. We have replaced “biomaterial constructs” with “biomaterials and scaffolds” in the revised manuscript.
Line 61: “…have not been widely examined.” What has been examined to date?
Author response: Thank you for the suggestion. We have corrected the sentence as below.” However, the application of CSPCs combined with biomaterials and scaffolds have been explored in cartilage regeneration in vitro [7], but the effects and biological behavior in osteochondral repair in vivo have not been widely examined.”
Line 74: “…CSPC-laden PLGA” Did you mean CSPC-loaded on PLGA?
Author response: Thank you for the suggestion. We have corrected “CSPC-laden PLGA” to “CSPC-loaded on PLGA” in the revised manuscript.
Lines 80-81: Please eliminate this sentence from the introduction.
Author response: Thank you for the suggestion. We have eliminated this sentence from the introduction.
MATERIALS AND METHODS
Lines 100-101: “After 20 min, non-adherent cells, namely osteoarthritis chondrocytes (OACs)…” Why OA chondrocytes? Previously it is written that articular cartilage samples were dissected from non-lesion surface areas of the knee joints. I can't understand the name of osteoarthritis chondrocytes.
Author response: Thank you for your question. We have mentioned that adult articular cartilage samples (53–90-year-old subjects; mean, 70 years; n = 16) were dissected from non-lesion surface areas of the knee joints of patients without signs of rheumatoid involvement undergoing total knee replacement surgery. The patient undergoing total knee replacement surgery which was in late stage of osteoarthritis (OA) and total knee replacement surgery is needed. Because there was lesion everywhere in cartilage surface in OA patient, we dissected the non-lesion surface areas to isolate the OACs and further purified CSPCs.
Line 106: Why did you first use FBS 20% and then 10%?
Author response: Thank you for the suggestion. We have corrected FBS to 10%.
Lines 119-120. “Adipogenesis was observed by detecting lipid droplets via Alizarin Red S staining and osteogenesis for mineralized bone matrix deposition by Oil Red O staining after 21 days.” It is the reverse, i.e. adipogenesis was observed by detecting lipid droplets by Oil Red O staining while osteogenesis for mineralized bone matrix was detected by Alizarin Red S.
Author response: Thank you for the suggestion. We have corrected the whole sentence in the revised manuscript.
Lines 122-129: Did you make micromasses in wells? Only by resuspending? No centrifuge?
Author response: Thank you for your question. We make micromasses without centrifugation because we make high-density micromass. We have added “high-density” in the revised manuscript.
Lines 132-141: The only way to reliably characterize MSCs through surface markers is for the cells to be cells negative for surface markers such as CD45, CD34, CD14 or CD11b, CD79 or CD19 and HLA-DR, but positive for CD105, CD73 and CD90.
Author response: Thank you for providing this thoughtful comment. CSPCs are endogenous chondroprogenitors located in the articular cartilage superficial zone in the joint cavity. All the joint-resident MSCs, such as synovium MSCs, synovial fluid MSC, infrapatellar fat pad MSC, or progenitor cells, possess the same surface marker, CD44 [1]. CD44 is also the cell surface receptor of hyaluronic acid (HA) [2], which is typically abundant in the joint cavity. This makes CD44 a marker for the joint-resident MSCs. In addition, we examined hematopoietic stem cell-associated markers, including CD34, CD45, and MSC-associated surface markers, such as CD90. Furthermore, we also examined CD146, a marker for CSPCs. CSPCs are tissue-specific progenitor cells located in the articular cartilage. In our study, to demonstrate that CSPCs maintain their original features similar to chondrocytes and osteocytes, we evaluated the expression of SOX9, RUNX2, and DCX. Our findings suggest that CSPCs have a regenerative potential and thus can treat osteochondral defects. Although we measured a limited number of MSC surface markers, the objective of study was not that of comparing CSPC with MSCs. To address the reviewer’s comment, we have included appropriate correction in the “The CSPCs harvested and sorted from patients with late-stage OA did not dedifferentiate like chondrocytes but expressed colony-forming ability and multilineage differentiation” section of the revised manuscript.
Lines 163-177:
-Did you do a power analysis to establish the number of animals per group?
Author response: Thank you for the suggestion. We did power analysis and followed the number of animals per group in previous study [3].
- Method of animal euthanasia?
Author response: Thank you for the question. The rabbits were euthanized after 4 or 12 weeks via intravenous injection of 2meq/kg KCL (Taiwan Biotech, Taoyuan, Taiwan). We have corrected the sentence in the revised manuscript.
-Please replace sacrificed to euthanized.
Author response: Thank you for the suggestion. We have replaced “sacrificed” to “euthanized” in the revised manuscript.
- Have you evaluated and removed the popliteal lymph nodes?
Author response: Thank you for your question. We did not remove the popliteal lymph nodes during surgery. After CSPC/PLGA implantation, we evaluated the rabbits with surgery at least twice a week, there are no redness, swelling around the knee joint in all rabbits. We have added “the sentence” in the revised manuscript.
Lines 186-192: why didn't you do also the trabecular thickness and separation.
Author response: We thank you for your question. Trabecular thickness (Tb.Th) and trabecular separation (Tb.Sp) were measured by locally inscribing a sphere of the maximal radius into the foreground (Tb.Th) and background (Tb.Sp) structures in 3D binary images [4]. Tb.Sp is measured using the same process as Tb.Th but is applied on the background phase of the input image. During the regeneration process, PLGA scaffold degrades first, followed by vascularization and then tissue ingrowth. In the Micro CT analysis, if the tissue has not grown back immediately after PLGA degradation, the cavity will be considered as Tb.Sp. The initial phase of the regeneration process are characterized by high level of Tb.Sp, whereas low level of Tb.Th are measured. Therefore, we used only Tb.Th to indicate the width of the bone growth. To explain this, we have added “the width of the bone growth as” in the revised manuscript.
RESULTS
Line 221: Figures 1A and B are of low resolution, and it is not possible to see what is described in text.
Author response: Thank you for your suggestion. We have enhanced the resolution in the pictures. There are many pictures in Figure 1, so some limitation existed. The better solution is that upload every single picture to get the best resolution.
Line 228: As for Figures 1A and B, also Figures B and C are of low resolution, and it is not possible to understand what is described in text.
Author response: Thank you for your suggestion. We have enhanced the resolution in the pictures. There are many pictures in Figure 1, so some limitation existed. The better solution is that upload every single picture to get the best resolution.
Line 240: Figure 1D are of low resolution and described in the text in a superficial way.
Author response: Thank you for your suggestion. We have enhanced the resolution of the pictures in the revised manuscript. There are many pictures in Figure 1, so some limitation existed. The better solution is that upload every single picture to get the best resolution.
Line 248. The immunophenotypes of CSPCs were poorly analyzed. What about ACAN, ADIPOQ, ALP, BGLAP, PPARG?
Author response: Thank you for the suggestion. In this study, we only investigated the effects of CSPCs combined with monophasic scaffold in osteochondral regeneration; therefore, we did not analyze ACAN, ADIPOQ, ALP, BGLAP, and PPARG. Collagen is the most common protein found in extracellular matrix, making up approximately 90% of the dry weight of articular cartilage [5, 6]. Therefore, we investigated type I collagen as an osteogenesis marker and type II collagen as a chondrogenesis marker using immunostaining during the regenerative process as reported in previous studies [7, 8].
Line 349: The histological images are too many, too small and with too low a magnification to evaluate what is reported in the text.
Author response: Thank you for the suggestion. We have enhanced the resolution in the images of the revised manuscript. The magnifications in images we used were 4x, 10x. Because using these magnifications we can observed morphology in cartilage and subchondral bone simultaneously
Line 374: The immunohistochemical images all look the same and the staining appears nonspecific. To evaluate what is reported in the text or in any case to detect differences, larger magnifications are required, i.e. 20x, 40x, 80x.
Author response: Thank you for the suggestion. The magnifications in immunohistochemical images we used were 4x, 10x. Because using these magnifications we can observed the type I and collage expressed in cartilage and subchondral bone simultaneously. We have enhanced the resolution of the images in the revised manuscript.
Figure 7 and 8: please report the magnifications for each image.
Author response: Thank you for the suggestion. We have reported the magnifications for each image in figure 7 and 8.
DISCUSSION
Please do not mention the figures in the discussion section.
Author response: Thank you for the suggestion. We have corrected this in the revised manuscript.
‘The CSPCs harvested and sorted from patients with late-stage OA did not dedifferentiate like chondrocytes and behaved similarly to MSCs in terms of colony-forming ability and multilineage differentiation’. The data obtained are too premature to state that CSPCs behaved similarly to MSCs. I think that CSPCs could represent a novel cell source for osteochondral regeneration, but supplementary studies and analyses are mandatory. Few differentiating factors and few surface markers were analyzed as well as too few immunohistochemical evaluations.
Unfortunately considering that many images are low resolution, the comments present in the discussion are difficult for me to understand and comment on.
I think that the discussion should be less peremptory, perhaps assuming specific CSPCs mechanisms rather than giving them as certain.
Author response: Thank you for the suggestion. We have enhanced the resolution of images. We do not have enough evidence to demonstrate that CSPCs behaved similarly to MSCs. We only demonstrated that CSPCs can be isolated and expanded to a sufficient number of cells in-vitro; therefore, when CSPCs combined with the monophasic biomaterial, the cells could still be recruited and regenerated the complex osteochondral tissue in-vivo. In previous attempts at medical interventions such as autologous chondrocyte implantation (ACI) [9], the chondrocytes were too few in number and dedifferentiated during expansion. Most importantly, ACI cannot produce functional cartilage. However, like chondrocytes, CSPCs are endogenous cells in cartilage, but with self-renewal, multilineage ability. Therefore, as CSPCs possess tissue specific characteristics similar to chondrocytes and osteoblasts, the growth factor is not needed. In this manner, we demonstrated that CSPCs combined with the PLGA scaffold biologically and functionally to regenerate the osteochondral tissue.
References
- Yang, Z.; Li, H.; Yuan, Z.; Fu, L.; Jiang, S.; Gao, C.; Wang, F.; Zha, K.; Tian, G.; Sun, Z., et al.Endogenous cell recruitment strategy for articular cartilage regeneration. Acta Biomaterialia 2020, 114, 31-52.
- Aruffo, A.; Stamenkovic, I.; Melnick, M.; Underhill, C. B.; Seed, B. CD44 is the principal cell surface receptor for hyaluronate. Cell 1990, 61, 1303-1313.
- Chang, N. J.; Lin, Y. T.; Lin, C. C.; Wang, H. C.; Hsu, H. C.; Yeh, M. L. The repair of full-thickness articular cartilage defect using intra-articular administration of N-acetyl-D-glucosamine in the rabbit knee: randomized controlled trial. Biomedical engineering online 2015, 14, 105.
- Reznikov, N.; Alsheghri, A. A.; Piché, N.; Gendron, M.; Desrosiers, C.; Morozova, I.; Sanchez Siles, J. M.; Gonzalez-Quevedo, D.; Tamimi, I.; Song, J., et al. Altered topological blueprint of trabecular bone associates with skeletal pathology in humans. Bone Reports 2020, 12, 100264.
- Maia, F. R.; Carvalho, M. R.; Oliveira, J. M.; Reis, R. L. Tissue Engineering Strategies for Osteochondral Repair. Adv Exp Med Biol 2018, 1059, 353-371.
- Yang, J.; Zhang, Y. S.; Yue, K.; Khademhosseini, A. Cell-laden hydrogels for osteochondral and cartilage tissue engineering. Acta Biomater 2017, 57, 1-25.
- Mendes, L. F.; Katagiri, H.; Tam, W. L.; Chai, Y. C.; Geris, L.; Roberts, S. J.; Luyten, F. P. Advancing osteochondral tissue engineering: bone morphogenetic protein, transforming growth factor, and fibroblast growth factor signaling drive ordered differentiation of periosteal cells resulting in stable cartilage and bone formation in vivo. Stem cell research & therapy 2018, 9, 42.
- An Autologous Bone Marrow Mesenchymal Stem Cell–Derived Extracellular Matrix Scaffold Applied with Bone Marrow Stimulation for Cartilage Repair. Tissue Engineering Part A 2014, 20, 2455-2462.
- Harris, J. D.; Siston, R. A.; Pan, X.; Flanigan, D. C. Autologous chondrocyte implantation: a systematic review. The Journal of bone and joint surgery. American volume 2010, 92, 2220-2233.

Reviewer 2 Report
A good paper of future clinical relevance.
Author Response
Author response: Thank you for your suggestion. I really appreciate your comment.

Reviewer 3 Report
This study investigates the progenitor cells and PLGA scaffold for induce osteochondral regeneration in rabbit knees. It is exactly similar to their previous study (ref 24), except this time the cells were extracted from human. However, it was implanted into same rabbit animal model. Maybe the manuscript is step up from previous work, but unfortunately, they have not compared the data and discuss it in comparison with previous work.
Following specific comments.
- Figure 3 is very similar to figure 1 of ref 24.
- What is the size of sodium chloride and what was size of porosities?
- The seeding density on the scaffold, why different from previous study (ref 24).
- Size of scaffold 3x3 mm2, too small, discuss translation to human or larger animal model.
- Discuss comparison with previous study (ref 24).
Author Response
Reviewer 3
Comments and Suggestions for Authors
This study investigates the progenitor cells and PLGA scaffold for induce osteochondral regeneration in rabbit knees. It is exactly similar to their previous study (ref 24), except this time the cells were extracted from human. However, it was implanted into same rabbit animal model. Maybe the manuscript is step up from previous work, but unfortunately, they have not compared the data and discuss it in comparison with previous work.
Following specific comments.
- Figure 3 is very similar to figure 1 of ref 24.
Author response: Thank you for your question. We used the same procedure to fabricate PLGA scaffold in these two studies. These pictures were similar but totally different, the parameters of SEM pictures are not the same.
- What is the size of sodium chloride and what was size of porosities?
Author response: Thank you for your question. The size of sodium chloride 300-500 μm in diameter and the size of porosities in scaffold is 300-500 μm. We have corrected the sentence in the revised manuscript.
- The seeding density on the scaffold, why different from previous study (ref 24).
Author response: We thank the reviewer for pointing out the issue. The reason for a discordant seeding density is that we used two different cell sources in these two studies. In our previous study (ref 24), we used endothelial progenitor cells with a seeding density of 5.0 × 105 cells/mL on the scaffold as described by Chang, N.J., et al. [1]. In the current study, the seeding density of the CSPCs was 5.0 × 106 cells/mL, as described in the article by Otto, I.A., et al. [2]. The scaffold size (diameter = 6 mm, height = 2 mm) described by the authors is almost three times bigger than the scaffold used in our current study (diameter = 3 mm, height = 3mm). Therefore, while Otto, I.A., et al. used 1.5 × 107 cells/mL as seeding density, we used 5.0 × 106 cells/mL. Moreover, because the two cell types possess different self-renewal and proliferation ability, it was more appropriate to use different seeding densities on the scaffold.
- Size of scaffold 3x3 mm2, too small, discuss translation to human or larger animal model.
Author response: We appreciate the reviewer for pointing this out. The size of PLGA scaffold can be increase by up to 6 mm in height and 6 mm in diameter when transferred in to the minipig model [3]. In this study, we showed that an acellular PLGA scaffold plus treadmill exercise could possess enough mechanical strength and further promote articular cartilage regeneration in minipigs. In contrast, the cellular PLGA scaffold with a critical size (diameter of 4 mm and height of 7 mm) resurface the cartilage in an ovine model of osteochondral focal defect [4]. These results revealed that the scaffold size of PLGA can be increased, and the system can be applied to big animal models too.
- Discuss comparison with previous study (ref 24).
Author response: We are grateful for the suggestion. In an investigation done previously (ref 24), we used continuous passive motion (CPM) to promote and maintain the chondrogenesis in EPCs loaded PLGA scaffolds during osteochondral regeneration [5]. I In the current study, CSPCs migrated and attracted additional cells to repair osteochondral tissue without other intervention in PLGA Scaffolds. However, a combination of CSPCs and EPCs with biphasic scaffold might benefit heterogeneous osteochondral tissue at early disease stage and encourage sufficient endogenous cell-based repair attempts. We have added this paragraph in discussion in the revised manuscript.
References:
- Chang, N. J.; Lam, C. F.; Lin, C. C.; Chen, W. L.; Li, C. F.; Lin, Y. T.; Yeh, M. L. Transplantation of autologous endothelial progenitor cells in porous PLGA scaffolds create a microenvironment for the regeneration of hyaline cartilage in rabbits. Osteoarthritis and cartilage 2013, 21, 1613-1622.
- Otto, I. A.; Levato, R.; Webb, W. R.; Khan, I. M.; Breugem, C. C.; Malda, J. Progenitor cells in auricular cartilage demonstrate cartilage-forming capacity in 3D hydrogel culture. European cells & materials 2018, 35, 132-150.
- Lin, C. C.; Chu, C. J.; Chou, P. H.; Liang, C. H.; Liang, P. I.; Chang, N. J. Beneficial Therapeutic Approach of Acellular PLGA Implants Coupled With Rehabilitation Exercise for Osteochondral Repair: A Proof of Concept Study in a Minipig Model. The American journal of sports medicine 2020, 48, 2796-2807.
- Caminal, M.; Peris, D.; Fonseca, C.; Barrachina, J.; Codina, D.; Rabanal, R. M.; Moll, X.; Morist, A.; Garcia, F.; Cairo, J. J., et al. Cartilage resurfacing potential of PLGA scaffolds loaded with autologous cells from cartilage, fat, and bone marrow in an ovine model of osteochondral focal defect. Cytotechnology 2016, 68, 907-919.
- Wang, H. C.; Lin, T. H.; Chang, N. J.; Hsu, H. C.; Yeh, M. L. Continuous Passive Motion Promotes and Maintains Chondrogenesis in Autologous Endothelial Progenitor Cell-Loaded Porous PLGA Scaffolds during Osteochondral Defect Repair in a Rabbit Model. Int J Mol Sci 2019, 20.

Round 2
Reviewer 3 Report
The paper has been improved and much better, although question regarding previous work in not fully discussed, but it is ok.
The following articles from Cells, are interesting and maybe should be referenced and discussed.
1: Monaco G, Ladner YD, El Haj AJ, Forsyth NR, Alini M, Stoddart MJ. Mesenchymal
Stromal Cell Differentiation for Generating Cartilage and Bone-Like Tissues In
Vitro. Cells. 2021 Aug 22;10(8):2165.
2: Hopkins T, Wright KT, Kuiper NJ, Roberts S, Jermin P, Gallacher P, Kuiper JH.
An In Vitro System to Study the Effect of Subchondral Bone Health on Articular
Cartilage Repair in Humans. Cells. 2021 Jul 27;10(8):1903.
3: Rim YA, Nam Y, Park N, Jung H, Lee K, Lee J, Ju JH. Chondrogenic
Differentiation from Induced Pluripotent Stem Cells Using Non-Viral Minicircle
Vectors. Cells. 2020 Mar 1;9(3):582.
